# Parent and Caregiver Perspectives towards Cannabidiol as a Treatment for Fragile X Syndrome

**DOI:** 10.3390/genes13091594

**Published:** 2022-09-06

**Authors:** Madison Maertens, Hailey Silver, Jayne Dixon Weber, Hillary Rosselot, Reymundo Lozano

**Affiliations:** 1Department of Genetics and Genomic Sciences, Icahn School of Medicine at Mount Sinai, New York, NY 10029, USA; 2Department of Counseling and Clinical Psychology, Teachers College, Columbia University, New York, NY 10027, USA; 3National Fragile X Foundation, McLean, VA 22102, USA; 4Department of Pediatrics, Icahn School of Medicine at Mount Sinai, New York, NY 10029, USA

**Keywords:** Cannabidiol, CBD, Fragile X Syndrome, caregiver perspectives, CBD treatment

## Abstract

Cannabidiol (CBD) is a non-intoxicating chemical in cannabis plants that is being investigated as a candidate for treatment in Fragile X Syndrome (FXS), a leading known cause of inherited intellectual developmental disability. Studies have shown that CBD can reduce symptoms such as anxiety, social avoidance, hyperactivity, aggression, and sleep problems. This is a qualitative study that utilized a voluntary-anonymous survey that consisted of questions regarding demographics, medical information, the form, type, brand, dose, and frequency of CBD use, the rationale for use, the perception of effects, side effects, and costs. The full survey contained a total of 34 questions, including multiple-choice, Likert-scale, and optional free-response questions. This research revealed that there are a wide range of types, brands, and doses of CBD being administered to individuals with FXS by their parents and caregivers. There were many reasons why CBD was chosen, the most common ones being that respondents had heard positive things about CBD from members of the community, the perception that CBD had fewer side effects than other medications, and because respondents felt that CBD was a more natural substance. Most of the parents and caregivers who responded agreed that CBD improved some of the symptoms of FXS and made a positive difference overall. CBD has the therapeutic potential to help relieve some FXS symptoms. Future research is necessary to understand the benefits of CBD in FXS.

## 1. Background

Fragile X Syndrome (FXS) is the leading known monogenetic cause of inherited intellectual developmental disorder (IDD) and autism spectrum disorder (ASD). FXS is caused by an expansion of the number of CGG repeats in the 5′UTR of the Fragile X messenger ribonucleoprotein 1 (*FMR1*) gene. An expansion of more than 200 CGG repeats indicates a full mutation, which leads to the methylation of the promoter and silencing of the *FMR1* gene, causing absent or low levels of the Fragile X messenger ribonucleoprotein (FMRP) [1,2]. FMRP regulates the translation of many messenger RNAs involved in synaptic plasticity [3]. The absence of FMRP may disrupt the balance of the endocannabinoid system. The two main endocannabinoid ligands are anandamide (AEA) and arachidonoyl glycerol (2-AG), and both are produced in postsynaptic cell membranes and modulate neurotransmission by binding to CB1 receptors [4]. In the *FMR1* knockout mice, a decrease of the production of 2-AG was described. In FXS, it is hypothesized that Cannabidiol (CBD) corrects the endocannabinoid system by increasing the levels of AEA and 2-AG [5,6,7]. Additional research is needed to further understand the role of FMRP in the human endocannabinoid pathway. In addition to acting directly through the endocannabinoids, CBD may also have therapeutic benefits through other mechanisms. The lack of FMRP in FXS has been associated with decreased GABA receptor expression [8]. CBD also acts as a positive allosteric modulator of GABA-A receptors, enhances GABA binding affinity, and may have additional anxiolytic effects through interactions with the serotonin 1A receptor [9,10]. 

In a study that included three individuals with FXS, CBD improved quality of life by reducing anxiety symptoms and improving their language skills, with no reported adverse events [11]. Another open-label study included 20 participants with FXS, ages 6–17 years, and aimed to assess the safety, tolerability, and efficacy of a transdermal CBD gel. CBD reduced anxiety and behavioral symptoms, and there was also a significant average reduction in manic/hyperactive behavior, social avoidance, general anxiety, compulsive behavior, irritability, stereotypy, and social unresponsiveness/lethargy [12]. The CBD gel was well tolerated, with no serious adverse events reported. Currently, a Phase 3 randomized, double-blind, placebo-controlled trial to assess the efficacy of the CBD transdermal gel is being conducted [13]. The clinical trials investigating the safety and efficacy of CBD as a candidate for alleviating FXS symptoms are essential. However, investigating the experiences and insights of the parents and caregivers who are independently giving over-the-counter CBD supplements to individuals with FXS is also valuable. Therefore, this study aimed to obtain information on why parents/caregivers choose to give CBD, and the brands, types, forms, and dosages. This study also investigated parent and caregiver perspectives regarding CBD efficacy and side effects.

## 2. Materials and Methods

The study population consisted of parents and caregivers of individuals with FXS who were receiving or had received CBD. The survey was distributed through the National Fragile X Foundation (NFXF) online post, as well as NFXF research recruitment emails that provided a link to the post. The survey was developed using the REDCap electronic data capture tools hosted by the Icahn School of Medicine at Mount Sinai Hospital [14,15] (Full survey available in Appendix A). The survey contained a total of 34 questions, comprising multiple-choice, Likert-scale, and optional free-response questions. Branching logic was used; if CBD supplements were not currently being used, caregivers answered a smaller subset of 11 questions, including questions about why CBD use was stopped or why CBD had not been given. This survey was developed after a detailed review of the literature. Additionally, a committee of FXS experts and experienced clinicians and researchers provided feedback to optimize the clarity of the survey, through the National Fragile X Foundation’s Research Readiness Program. The study was exempted by the Mount Sinai Program for the Protection of Human Subjects Institutional Review Board (IRB-20-04047). The data was collected through REDCap, and included demographic information, medical information, the form, type, brand, dose, and frequency of CBD use, the rationale for use, the perception of effects and side effects, and financial information. Microsoft Excel was used to perform descriptive statistics of categorical variables, which are presented as frequencies. Data was presented using tables and graphs.

## 3. Results

A total of 25 caregivers completed the electronic survey. Of those, 15/25 (60%) indicated that the person with FXS that they cared for was currently taking CBD supplements. The other 10/25 (40%) caregivers indicated that the person with FXS either had taken CBD in the past, but stopped, or had not taken CBD before, although they had considered giving it. Demographic information from the 25 individuals with FXS was collected (Table 1). The majority of individuals were White (20/25, 80%), male (22/25, 88%), and between the ages of 5 and 29 years old (17/25, 68%). Of the 15 individuals who were currently taking CBD, all 15/15 (100%) had intellectual developmental disability (IDD), mostly moderate to severe (14/15, 93%), anxiety, mostly moderate to severe (13/15, 87%), attention problems, mostly moderate to severe (14/15, 93%), and some degree of hypersensitivity. Some individuals also had mild or moderate irritability (13/15, 87%), mild or severe seizures (3/15, 20%), mild to moderate ASD (8/15, 53%), mild to moderate aggression (12/15, 80%), mild self-injury (9/15, 60%), and mild to severe sleep problems/disorders (9/15, 60%) (Table 2). Most individuals were taking medications for hyperactivity, anxiety, and/or aggression. 

The types, forms, and brands of CBD that were used all varied greatly; Most caregivers reported using either pure CBD supplements (5/15, 33%) or full spectrum CBD supplements (6/15, 40%) where the supplement contains CBD as well as all of the other compounds that naturally occur in cannabis plants, including small traces of THC (less than 0.3 percent). Only 2/15 (13%) respondents indicated that they gave broad spectrum CBD, which is a supplement that contains CBD and other natural cannabis plant compounds, though any trace of THC is removed. The compound type was unknown for 2/15 (12%). Forms of CBD used included CBD oils or tinctures (3/15, 20%), CBD gummies or edibles (4/15, 27%), CBD capsules or pills (4/15, 27%), and CBD topical form (2/15, 13%). There were many different brands used, and only three individuals were using the same brand. For frequency of use, the majority (10/15, 67%) of caregivers reported that the individual had been taking CBD for about 1–2 years, whereas 2/15 (13%) reported 3–5 years, 1/15 (7%) reported 1–6 months, and 2/15 (13%) reported less than 1 month. A total of 13/15 (87%) caregivers reported that individuals with FXS were given CBD every day; however, one caregiver (1/15, 7%) reported that an individual was given CBD 2–4 times a week, and one reported that an individual was given CBD less than once a week. On the days that the individual was given CBD, 8/15 (53%) respondents reported giving the person CBD twice per day, 5/15 (33%) reported that they gave it only once per day, one respondent (1/15, 7%) reported giving it three times per day, and one (1/15, 7%) reported giving it 3–4 times per day. There was a wide range of doses of CBD given, ranging from 10 mg to 275 mg per day (Table 3). We listed the ages of each individual with FXS corresponding to the dose of CBD that they were given per day to assess if younger individuals were receiving lower doses and if older individuals were receiving higher doses, but this was not the case. It was noted that the individual receiving 275 mg per day was using CBD for seizure control.

Respondents reported which symptoms they aimed to treat using the CBD supplements; The top six symptoms that caregivers reportedly wanted to treat using CBD included anxiety (13/15, 87%), irritability (10/15, 67%), attention problems (6/15, 40%), hypersensitivity (6/15, 40%), sleep problems and disorders (5/15, 33%), and aggression (5/15, 33%). The respondents indicated how much they agreed or disagreed with the following reasons, “other treatments or medications have not been very effective”, “that they heard positive things about CBD from a medical provider, or a friend or someone in the community”, and that “CBD is a more natural option than prescribed drugs”. The majority of respondents agreed with these four statements. Out of these four, the reason the respondents agreed with the most was that they had heard positive things from a friend or someone in the community, with which 8/15 (53%) respondents strongly agreed and 5/15 (33%) agreed. A total of 10/15 (67%) agreed that they administered CBD because they could obtain it without a prescription. Another primary reason was that CBD had fewer side effects than other medications, with which 6/15 (40%) respondents strongly agreed. Lastly, when asked how much respondents agreed or disagreed with the two statements, “I do not trust FDA approved medications” and “I do not trust pharmaceutical companies”, 10/15 (67%) respondents disagreed or strongly disagreed with these statements.

The respondents also reported which changes in symptoms they observed with CBD use, most of which were improvements (Figure 1). Anxiety had been the primary reported symptom that they wanted to treat; 2/15 (13%) respondents indicated that the individual’s anxiety was very much improved during CBD use, and 6/15 (40%) indicated that anxiety was much improved. Furthermore, anxiety symptoms showed the most improvements, with 12/15 (80%) respondents having stated that there was at least some degree of improvement in the individual’s anxiety. 

Regarding attention and hypersensitivity, 4/15 (27%) respondents indicated that these problems were much improved, and over 9/15 (60%) that there was at least minimal improvement. However, one (1/15, 7%) reported that attention problems were minimally worse. Most respondents indicated that aggression, irritability, and sleep problems also improved to some degree during CBD use. No one reported that any of the other aforementioned symptoms had worsened. Caregivers also reported any changes they observed in behaviors, such as screaming or yelling inappropriately, temper tantrums, and others (Figure 2). Most respondents perceived some degree of improvement in irritability and whining, with 3/15 (20%) indicating that this behavior was much improved and 1/15 (7%) indicating that it was very much improved. Most respondents also saw improvements in temper tantrums and outbursts (9/15, 60%). However, 1/15 (7%) indicated that irritability and wining was minimally worse. Some respondents also observed much or very much improvement in the behaviors, “seeking isolation” and “withdrawn/preferring solitary activities”. 

Parents and caregivers were asked to indicate how much the CBD supplements had made a difference overall in the individual’s symptoms, and 7/15 (47%) reported much improved and 3/15 (20%) very much improved (Table 4). Feelings about stopping the CBD were also explored (Table 4). 

Most parents and caregivers reported that side effects such as restlessness, irritability, sleepiness, sleep problems, paranoia, psychosis, or change in appetite or weight were not experienced with the use of CBD; only one (1/15, 7%) reported moderate irritability, and another (1/15, 7%) reported moderate restlessness. Information about parents’ perception about the cost of over-the-counter (OTC) CBD supplements and how much they were spending on CBD was collected. A total of 7/15 (47%) respondents reported spending less than $50 per month, 3/15 (20%) reported spending between $50 and $100 per month, 1/15 (7%) reported spending up to $200 per month, 2/15 (13%) reported spending as much as $200–$300 per month, 1/15 (7%) reported spending up to $500 per month, and 1/15 (7%) reported spending more than $500 per month. Fourteen caregivers responded when asked how they felt about the cost of CBD, of which 4/14 (29%) reported feeling that the cost of OTC CBD was unreasonable, and 4/14 (29%) reported feeling that the cost was reasonable. A total of 2/14 (14%) reported feeling neutral about the cost of CBD, and the last 4/14 (29%) selected “other”. One of the respondents who selected “other” explained that they felt that the cost was “worth it”, and another respondent who selected “other” explained that they felt that OTC options are “very expensive” for the dose that is needed. The final question of the electronic survey was open response, allowing parents and caregivers to share any other experiences they had from using CBD to treat FXS. One of the caregivers wrote, “He is so much better on this than any other med we have ever tried… We will not go without it”. Another wrote, “I thought it might decrease anxiety and therefore improve school performance. However, no improvement in schoolwork”. Another explained, “CBD was life changing for our child, he had intractable seizures that were worsening over the years… We noticed the seizures diminishing within the 1st month of using CBD… Our son has been seizure free for almost 5 years”. A total of 10/25 (40%) caregivers reported that the person with FXS was not currently taking CBD supplements, and 7 of those 10 reported that they had only considered giving CBD. Regarding the reasons CBD was not given, 5/7 (71%) chose “I do not know enough about CBD, so I was not comfortable giving it”, 3/7 (43%) chose, “I did not think CBD would help with the FXS symptoms”, 1/7 (14%) indicated “I don’t know where to get CBD”, and 1/7 (14%) chose “CBD is too expensive”. The remaining 3 of the 10 reported that they had given CBD in the past and stopped because of side effects and/or an absence of benefits. 

## 4. Discussion

The limitations of this research include the small sample size, lack of ethnic and sex diversity, and the inability to assess other medications or interventions as confounding variables. Additionally, this was an anonymous survey collecting subjective self-reported information, which can lead to recall bias. This survey revealed that there is a very wide range of types, brands, and doses of CBD that are being used; therefore, it is difficult to analyze the efficacy of CBD. Additionally, these results emphasize the numerous brands and dosages available for CBD consumers with many different concentrations and forms. The respondents had heard positive things from a friend or someone in the community about CBD as a treatment for FXS, and not from physicians. Obtaining this information from the families is important and can help physicians be aware of the dosages used in the community and the benefits. Some of the primary symptoms that caregivers reported that the individuals with FXS experienced before CBD supplement use included anxiety, irritability, attention problems, hypersensitivity, sleep problems, and aggression. Anxiety was the primary symptom that parents wanted to treat, which is not surprising given that anxiety is very prominent in FXS, and CBD is used as an anxiolytic. Most of the parents and caregivers agreed that CBD supplements improved at least some of the FXS symptoms, including but not limited to anxiety, irritability, and hypersensitivity. This finding also reinforces the theory that CBD may have more significant, positive, global effects on behavioral and emotional symptoms; however, further research is necessary. 

Overall, caregivers generally reported that CBD made a significant and positive difference to individuals’ symptoms. Most parents reported that side effects were not experienced, and any that were reported were generally mild or very mild. Lastly, the cost of OTC CBD can be challenging or expensive for some, and a few parents reported that the cost was unreasonable. Therefore, CBD may currently be more accessible to families with higher incomes. Placebo-controlled clinical trials regarding CBD as a treatment for FXS are essential to better understanding the efficacy and safety of CBD in FXS. It is also important to study caregiver and family perspectives on the impact CBD has on daily functioning. Incorporating caregiver surveys during CBD clinical trials may also provide additional valuable insights.

## Figures and Tables

**Figure 1 genes-13-01594-f001:**
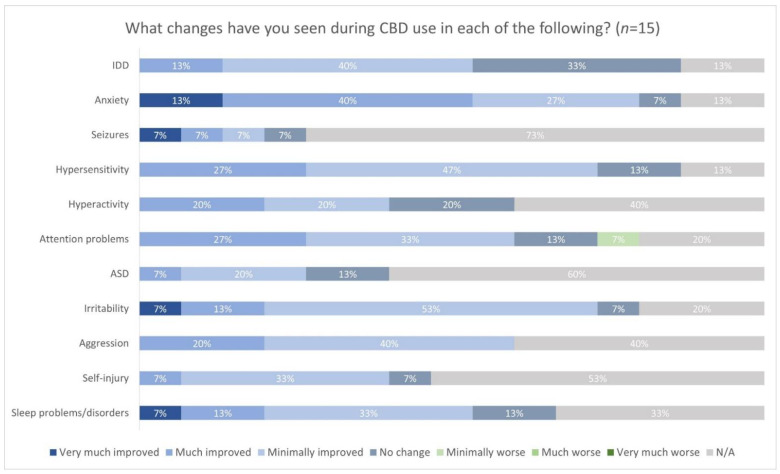
Changes seen with the use of CBD. Footnote: Intellectual Developmental Disability (IDD); Autism Spectrum Disorder (ASD).

**Figure 2 genes-13-01594-f002:**
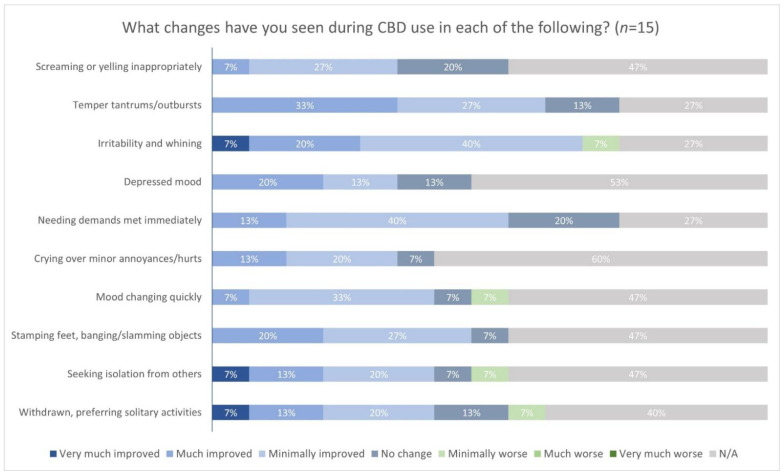
Changes in behaviors with the use of CBD. Footnote: Intellectual Developmental Disability (IDD); Autism Spectrum Disorder (ASD).

**Table 1 genes-13-01594-t001:** Demographic characteristics of individuals with FXS (*n* = 25).

	Demographic Characteristics				
Variable	N	%	Variable	Median	Range (Min, Max)
Ethnicity			Age	20	(5.57)
White	20	80%			
Latino/Hispanic	1	4%			
Asian	2	8%			
White/Asian	1	4%			
White/African American	1	4%			
Sex					
Male	22	88%			
Female	3	12%			

**Table 2 genes-13-01594-t002:** Severity of signs and symptoms experienced by individuals with FXS who were using CBD (*n* = 15).

Sign/Symptom		Severity			Total
	Very Mild	Mild	Moderate	Severe	Present
IDD	1 (7%)	0 (0%)	10 (67%)	4 (27%)	15 (100%)
Anxiety	0 (0%)	2 (13%)	12 (80%)	1 (7%)	15 (100%)
Seizures	0 (0%)	1 (7%)	0 (0%)	2 (13%)	3 (20%)
Hypersensitivity	4 (27%)	4 (27%)	6 (40%)	1 (7%)	15 (100%)
Hyperactivity	4 (27%)	4 (27%)	3 (20%)	1 (7%)	12 (80%)
Attention problems	0 (0%)	1 (7%)	11 (73%)	3 (20%)	15 (100%)
ASD	3 (20%)	2 (13%)	3 (20%)	0 (0%)	8 (53%)
Irritability	0 (0%)	4 (27%)	9 (60%)	0 (0%)	13 (87%)
Aggression	6 (40%)	2 (13%)	4 (27%)	0 (0%)	12 (80%)
Self-injury	5 (33%)	4 (27%)	0 (0%)	0 (0%)	9 (60%)
Sleep problems	4 (27%)	2 (13%)	1 (7%)	2 (13%)	9 (60%)

Intellectual Developmental Disability (IDD); Autism Spectrum Disorder (ASD).

**Table 3 genes-13-01594-t003:** Daily doses of CBD (mg/day) and ages of individuals with FXS (n = 10).

Dose Given	Age
10 mg/day	19
10 mg/day	16
10 mg/day	15
14 mg/day	7
16 mg/day	46
25–50 mg/day	32
30 mg/day	34
100 mg/day	41
240 mg/day	10
275 mg/day	14

The daily dose of CBD was only available for 10 of 15 individuals.

**Table 4 genes-13-01594-t004:** Overall improvement from CBD of symptoms and how caregivers would feel if they had to stop CBD use.

	Total (n)	%
**How much CBD has improved symptoms?**		
Very much improved	3	20%
Much improved	7	47%
Minimally improved	3	20%
Neutral	2	13%
Minimally worse	0	0%
Much worse	0	0%
Very much worse	0	0%
**Feelings if parents had to stop CBD Use**		
Not disappointed at all	2	13%
Slightly disappointed	2	13%
Moderately disappointed	2	13%
Very disappointed	6	40%
Extremely disappointed	3	20%

## Data Availability

Data of the full survey is available upon request.

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
