# Peer review of "Parent and Caregiver Perspectives towards Cannabidiol as a Treatment for Fragile X Syndrome"

_genes, 2022, doi:10.3390/genes13091594_

Round 1

Reviewer 1 Report

The study result useful, fluent and interesting to read, although some details should be improved.

-The introduction could be implemented about the role of FMRP, in particular in its role in the endocannabinoid pathway, considering that mouse model not always reproduce exactly the FXS in humans.

-About results, from line 101 is reported 5 caregiver use pure CBD supplemets, 6 use full spectrum CBD, 2 gave broad spectrum CBD, what about the other 2? Because the pool is represented by 15 individuals.

The same about the forms of CBD used, the frequency and the table 1. What about the other missing individuals?

As mentioned from the authors in the discussion, the critical points of the study are represented from three points. The small sample size and the variety of frequency, doses and brands that caregivers uses with FXS individuals. Could be useful working to identify the best conditions for the efficancy of CBD, reducing heterogeneity. For all these reasons, is not possible considering this study as quantitative. Moreover, no one quantitative or objective scale was used to quantify the levels of improvements (mentioned in Fig1) and abnormal behaviours (mentioned in Fig2). Every evaluation is assessed from the caregivers, resulting subjective.  It is appropriate to consider this work as qualitative and not quantitative, removing "quantitative" from the line 15 in the abstract.

Author Response

The study result useful, fluent and interesting to read, although some details should be improved.

-The introduction could be implemented about the role of FMRP, in particular in its role in the endocannabinoid pathway, considering that mouse model not always reproduce exactly the FXS in humans.

We agree, and at this time FMRP’s role in humans’ endocannabinoid pathway is not fully understood. Therefore, we added (Line 44), “Additional research is needed to further understand the role of FMRP in the human endocannabinoid pathway” to the introduction.

-About results, from line 101 is reported 5 caregiver use pure CBD supplemets, 6 use full spectrum CBD, 2 gave broad spectrum CBD, what about the other 2? Because the pool is represented by 15 individuals.

Since the compound type was unknown for 2 of the 15, the following sentence was added to line 121 (formerly line 107 before revisions): “The compound type was unknown for 2/15 (12%).”

The same about the forms of CBD used, the frequency and the table 1. What about the other missing individuals?

Thank you for your question. At line 109 to include the forms of CBD used by the two missed respondents, the following was added, “and CBD topical forms (2/15, 13%)”. Since the respondents were not required to answer every item on the questionnaire, unfortunately we did not have answers to some of the dose and frequency items. Therefore, in some of the results discussion and in table 1 we were only able to compare the daily doses and ages for 10 of the 15 respondents. A foot note was added to explain this. This table is now re-assigned table 3, and a new table 1 was added with demographic information for all 25 individuals.

As mentioned from the authors in the discussion, the critical points of the study are represented from three points. The small sample size and the variety of frequency, doses and brands that caregivers uses with FXS individuals. Could be useful working to identify the best conditions for the efficancy of CBD, reducing heterogeneity. For all these reasons, is not possible considering this study as quantitative. Moreover, no one quantitative or objective scale was used to quantify the levels of improvements (mentioned in Fig1) and abnormal behaviours (mentioned in Fig2). Every evaluation is assessed from the caregivers, resulting subjective.  It is appropriate to consider this work as qualitative and not quantitative, removing "quantitative" from the line 15 in the abstract.

Thank you for your explanation, “quantitative” from the line 15 in the abstract has been changed to “qualitative”.

Reviewer 2 Report

This is very interesting and useful study which provide important data for medical professionals in the fragile X field as well as for families worldwide. The study revealed that there is a very wide range of types, brands, and doses of CBD that are being used. Also, these results emphasize the numerous brands and dosages available for CBD consumers with many different concentrations and forms. Some of the primary symptoms that caregivers reported before providing CBD supplements included anxiety, irritability, attention problems, hypersensitivity, sleep problems, and aggression. Anxiety was the symptoms that most parents wanted to treat, this is not surprising giving the fact that anxiety is very prominent in FXS and that CBD is used as anxiolytic. Most of the parents and caregivers agreed that CBD supplements improved at least some of the FXS symptoms, including but not limited to anxiety, irritability, and hypersensitivity. 

However, I have a few minor suggestion and questions.

1. Introduction, line 38: 2-archidonoyl glycerol (2-AG) is missed. There is only abbreviation without full name of endocannabinoid ligand. Please, delete the full name of 2-AG in line 41, and move it in line 38 (the first appearing of this ligand).

2. Methods: It can be useful if you can describe statistical methods that were used in the study.

3. Results: is there any possibility to presents results from line 87 to line 98 in separate table? It will be more transparent. 

4. Current Table 1 presents daily doses of CBD (mg/day) and ages of the individuals with FXS. However, there are only 10 rows for 10 individuals. Please, can you add explanation as Table foot note  why are only 10 individuals presenting in this table? The 15 individuals were currently taking CBD.

5. In addition, the same table presents different ages: from 7 to 46 y. However, these data are not consistent with the age data in line 92.  It will be clearer if age of whole group presented in methods, with average age, range and Median value. 

6. Ten (40%) caregivers reported that the person with FXS was not currently taking CBD supplements. The percentages presented in line 193-198 are not clear. 

Overall, my main suggestion is to present all frequencies more clearly. For example, present frequencies as 10/25 (40%).....

Author Response

 We appreciate the thoughtful comments of reviewer 2, please see our responses to the comments below.  

I have a few minor suggestion and questions.

Cover Letter Detailing Revision Changes - Reviewer 2

This is very interesting and useful study which provide important data for medical professionals in the fragile X field as well as for families worldwide. The study revealed that there is a very wide range of types, brands, and doses of CBD that are being used. Also, these results emphasize the numerous brands and dosages available for CBD consumers with many different concentrations and forms. Some of the primary symptoms that caregivers reported before providing CBD supplements included anxiety, irritability, attention problems, hypersensitivity, sleep problems, and aggression. Anxiety was the symptoms that most parents wanted to treat, this is not surprising giving the fact that anxiety is very prominent in FXS and that CBD is used as anxiolytic. Most of the parents and caregivers agreed that CBD supplements improved at least some of the FXS symptoms, including but not limited to anxiety, irritability, and hypersensitivity. 

However, I have a few minor suggestion and questions.

  1. Introduction, line 38: 2-archidonoyl glycerol (2-AG) is missed. There is only abbreviation without full name of endocannabinoid ligand. Please, delete the full name of 2-AG in line 41, and move it in line 38 (the first appearing of this ligand).

Thank you for catching this error, the full name (2-archidonoyl glycerol) was deleted from line 42 (previously line 41 before revisions) and moved to line 39 (previously line 38 before revisions) where the ligand first appeared.

  1. Methods: It can be useful if you can describe statistical methods that were used in the study.

The following was added to the end of the methods section: “Microsoft Excel was used to perform descriptive statistics of categorical variables, which are presented as frequencies. Data was presented using tables and graphs.”

  1. Results: is there any possibility to presents results from line 87 to line 98 in separate table? It will be more transparent. 

A new table 1 (now table one and the previous table one was re-assigned table 3) was added with demographic information from this referenced section for all 25 individuals. A new table 2 was also added with the sign/symptom information from this referenced section.

  1. Current Table 1 presents daily doses of CBD (mg/day) and ages of the individuals with FXS. However, there are only 10 rows for 10 individuals. Please, can you add explanation as Table foot note why are only 10 individuals presenting in this table? The 15 individuals were currently taking CBD.

Thank you for your question. Since the respondents were not required to answer every item on the questionnaire, unfortunately we did not have answers to some of the dose and frequency items. Therefore, in table 1 we were only able to compare the daily doses and ages for 10 of the 15 respondents. A foot note was added under Table1 to explain this.

  1. In addition, the same table presents different ages: from 7 to 46 y. However, these data are not consistent with the age data in line 92.  It will be clearer if age of whole group presented in methods, with average age, range and Median value.

Thank you, I am happy to explain this difference/inconsistency. The age data in line 92 represents all 25 responses, including those that reported currently giving CBD and those that reported not currently giving CBD. Conversely, Table 1 is only including the ages of those that were currently taking CBD and also provided daily dose information. Additionally, this table is now re-assigned table 3, and a new table 1 was added with demographic information for all 25 individuals.

  1. Ten (40%) caregivers reported that the person with FXS was not currently taking CBD supplements. The percentages presented in line 193-198 are not clear. 

The frequencies in lines 218 (formally lines 193-198 before revisions) have been revised so that they are clearer and in a uniform format.

Overall, my main suggestion is to present all frequencies more clearly. For example, present frequencies as 10/25 (40%).....

The frequencies have been revised to the above format, fraction (percentage), as suggested to improve clarity.

Reviewer 3 Report

In this manuscript the authors have described the survey results from parents and caregivers of individuals with fragile X syndrome and their experiences with cannabidiol as a treatment. The paper is a succinct and straight-forward summary of the experiences reported from the 25 respondents to the online survey that was administered. I only have a few small suggested edits.

For Figure 1, it may make more sense to present the data for each group without the N/A category and include the total sample size for each group.

Line 15 – “…study that utilized…” instead of “…study utilized…”

Line 18 – delete “few” from “… and few optional free-response questions.”

Line 38 – the first time 2-AG is mentioned, it should be spelled out as it is in line 40-41.

Line 52 – “years” instead of “year”

Author Response

We thank the reviewer for taking the time to review our manuscript and for the comments. 

Cover Letter Detailing Revision Changes - Reviewer 3

In this manuscript the authors have described the survey results from parents and caregivers of individuals with fragile X syndrome and their experiences with cannabidiol as a treatment. The paper is a succinct and straight-forward summary of the experiences reported from the 25 respondents to the online survey that was administered. I only have a few small suggested edits.

For Figure 1, it may make more sense to present the data for each group without the N/A category and include the total sample size for each group.

We decided to include the patients that selected N/A for transparency, to include all 15 of the participants, and to avoid inflation of the percentages in improvements. This was also the recommendation of reviewer 2. 

Line 15 – “…study that utilized…” instead of “…study utilized…”

Thank you for catching this error, at line 15 this phrase was corrected to “study that utilized”.

Line 19 (formally line 18 before revision) – delete “few” from “… and few optional free-response questions.”

“Few” was deleted from line 18, thank you.

Line 38 – the first time 2-AG is mentioned, it should be spelled out as it is in line 40-41.

The full name was moved to the correct place, where the ligand was first mentioned.

Line 52 – “years” instead of “year”

At line 55 (formally line 52 before revisions), “Year” was corrected to “years”.